# Dietary Branched-Chain Amino Acids (BCAAs) and Risk of Dyslipidemia in a Chinese Population

**DOI:** 10.3390/nu14091824

**Published:** 2022-04-27

**Authors:** Lianlong Yu, Qianrang Zhu, Yuqian Li, Pengkun Song, Jian Zhang

**Affiliations:** 1National Institute for Nutrition and Health, Chinese Center for Disease Control and Prevention, Beijing 100050, China; lianlong00a@163.com (L.Y.); zhuqianrang@hotmail.com (Q.Z.); cnu_lyq@126.com (Y.L.); songpk@ninh.chinacdc.cn (P.S.); 2NHC Key Laboratory of Trace Element Nutrition, National Institute for Nutrition and Health, Chinese Center for Disease Control and Prevention, Beijing 100050, China

**Keywords:** branched-chain amino acids, dyslipidemia, blood lipids, case–control study

## Abstract

This study aimed to explore the association between dietary BCAAs, blood lipid levels and risk of dyslipidemia. In this case–control study, a total of 9541 subjects with normal blood lipids were included as a control group, and 9792 patients with dyslipidemia were included as a case group. Dietary BCAA intake data were measured using 3-day 24 h meal recalls and household condiment weighing. All samples were from China Nutrition and Health Surveillance (2015). Generalized linear model, logistic regression, and restricted cubic spline (RCS) were used to evaluate the relationship between dietary BCAAs, blood lipids and dyslipidemia. After adjusting for confounding factors, dietary BCAAs were positively correlated with TC and LDL-C (*p* < 0.05). Higher dietary BCAAs were associated with higher OR for Hypercholesteremia (Q4 vs. Q1, OR = 1.29, 95% CI: 1.05–1.58, *p*-trend = 0.034). The ORs of Hyper-LDL-cholesterolemia showed inverted U-shaped with increasing dietary BCAAs (Q3 vs. Q1, OR = 1.20, 95% CI: 1.03–1.39; Q2 vs. Q1, OR = 1.05, 95% CI: 1.01–1.31). The relationship between dietary BCAAs and the risk of Hypercholesteremia and Hyper-LDL-cholesterolemia were both nonlinear (*p* nonlinearity = 0.0059, 0.0198). Our study reveals that dietary BCAAs are associated with specific types of lipids and risk of dyslipidemia, some of which may be non-linear.

## 1. Instruction

Dyslipidemia is characterized by elevated total cholesterol (TC) or triglycerides (TG) or low-density lipoprotein cholesterol (LDL-C), or decreased high-density lipoprotein cholesterol (HDL-C) [1]. The overall prevalence of dyslipidemia in Chinese adults was 34.0% (2007–2010) [2], The prevalence of Hypercholesteremia, Hyper-LDL-choles terolemia, Hypo-HDL-cholesterolemia and Hypertriglyceridemia were 6.9%, 8.1%, 20.4% and 13.8%, respectively, in China (2013–2014) [3]. Unfortunately, dyslipidemia is the key risk factor for atherosclerosis and cardiovascular disease (CVD) [4,5]. Currently, the worldwide prevalence of dyslipidemia has become a public health problem worthy of attention [6,7,8], which has also caused a huge economic burden [9]. Fortunately, previous studies have shown that there is a close relationship between dyslipidemia and diet [10,11,12], and dyslipidemia can be prevented and improved through diet [13]. Therefore, exploring the risk factors of dyslipidemia among dietary factors is of great significance for the prevention of CVD.

Branched-chain amino acids (BCAAs) are an important component of dietary protein, including leucine, isoleucine and valine. They are all essential amino acids [14]. Since BCAAs cannot be synthesized in the human body, the BCAAs must be obtained from food [15]. Past studies have shown that BCAA supplementation has a positive effect on preventing muscle loss and improving muscle function [16,17]. However, a large number of studies in recent years have found that high levels of dietary or serum BCAAs are risk factors for certain chronic diseases, such as obesity, insulin resistance and diabetes [18,19,20]. Meanwhile, studies have shown that restricting the intake of BCAAs can prolong lifespan [21]. In addition, serum BCAAs have been shown to be a biomarker for dyslipidemia [22,23]. A study from Finland [24] demonstrated that high concentrations of BCAAs in circle blood were significantly associated with lipid metabolism.

However, the current research about the effects of dietary BCAAs on blood lipids and dyslipidemia is very limited. The clarification of this relationship has important implications for the prevention of dyslipidemia through diet. Therefore, we conducted this case–control study in a Chinese population to explore the relationship between dietary BCAAs, dyslipidemia risk and lipid levels.

## 2. Research Design and Methods

### 2.1. Study Population and Data Collection

A total of 19,333 participants were included in this study: 9541 subjects with normolipidemia as a control group, 9792 subjects with dyslipidemia as a case group, including 2539 subjects with Hypercholesteremia, 4523 subjects with Hypertriglyceridemia, 5461 subjects with Hypo-HDL-cholesterolemia and 2700 subjects with Hyper-LDL-cholesterolemia. Among the dyslipidemia patients, 179 subjects had 4 kinds of dyslipidemia simultaneously; 204 subjects had Hypercholesteremia, Hypertriglyceridemia and Hypo-HDL-cholesterolemia; 450 subjects had Hypercholesteremia, Hypertriglyceridemia and Hyper-LDL-cholesterolemia; 144 subjects had Hypercholesteremia and Hypertriglyceridemia; 47 subjects had Hypercholesteremia, Hypo-HDL-cholesterolemia and Hyper-LDL-cholesterolemia; 19 subjects had Hypercholesteremia and Hypo-HDL-cholesterolemia; 1192 subjects had Hypercholesteremia and Hyper-LDL-cholesterolemia; 112 subjects had Hypertriglyceridemia, Hypo-HDL-cholesterolemia and Hyper- LDL-cholesterolemia; 1753 subjects had Hypertriglyceridemia and Hypo-HDL-cholesterolemia; 148 subjects had Hypertriglyceridemia and Hyper-LDL-cholesterolemia; 69 subjects had Hypo-HDL-cholesterolemia and Hyper-LDL-cholesterolemia (Figure 1). All cases were from China Nutrition and Health Surveillance (2015) (CHNS 2015). This survey is a nationwide monitoring of residents’ nutritional status conducted by the Chinese Center for Disease Control and Prevention (China CDC), aiming to evaluate residents’ nutritional status, health and disease conditions and living habits. This project covered 31 provinces in China, including a total of 302 monitoring districts and counties. The specific details of this project are described in the previous literature [25]. In this case–control study, cases with dyslipidemia were frequency-matched with control subjects based on sex, age (±2 years) and BMI (±3 kg/m^2^). Inclusion criteria for all subjects were Age ≥ 30 years old, BMI < 40 kg/m^2^. The inclusion criteria of control group were no history of dyslipidemia diagnosis, and normal lipid level in clinical examination and blood tests during this survey. All subjects suffering from hypertension, diabetes, coronary heart disease, stroke, chronic obstructive pulmonary disease, asthma, bone and joint disease, neck and shoulder disease, chronic digestive system disease, chronic urinary system disease, tumor and other diseases were excluded from the study.

The diagnostic criteria for dyslipidemia refer to the “Guidelines for the Prevention and Treatment of Dyslipidemia in Chinese Adults (Revised 2016)” [26], that is, Hypercholesteremia: TC ≥ 6.20 mmol/L; Hypercholesteremia: TG ≥ 2.30 mmol/L; Hyper-LDL-cholesterolemia: LDL-C ≥ 4.10 mmol/L; Hypo-HDL-cholesterolemia: HDL-C < 1.00 mmol/L, any type of them was called dyslipidemia. Basic characteristics, including age, gender, body mass index (BMI), blood pressure, metabolic equivalent (MET-h/d), smoking status, drinking status and education level, were obtained through standardized questionnaires designed by national survey groups. Anthropometric measurements, such as weight (kg) and height (cm), were conducted by uniformly trained staff of district/county CDC. All subjects underwent physical examination in the early morning after an overnight fast, and subjects were instructed to remove their shoes and heavy clothing and untie their hair buns during the measurement. BMI was calculated after the survey, and the formula was weight (kg)/square of height (m^2^). At the same time, the equipment used for body measurement were the unified model, electronic weight scale (TANITA HD-390) and electronic blood pressure monitor (OMRON HBP1300), and the measurement accuracy is 0.1 cm, 0.1 kg and 1 mmHg, respectively. This project and study were approved by the Ethics Committee of the Chinese Center for Disease Control and Prevention (approval number: 201519-B). At the same time, all participants signed an informed consent form before participating in the project.

### 2.2. Dietary BCAAs Assessment

Dietary data were collected using the 24 h retrospective food recall method to investigate the diet of each subject for three days (including two weekdays and one weekend), while the intake of condiments such as oil and salt was measured using a three-day household weighing method. Investigators are staff from district/county CDCs and community hospitals, who have received strict and uniform training on dietary surveys, and are trained by the national and provincial CDCs. During the investigation, subjects were asked to maintain their previous eating and living habits. Dietary energy, carbohydrate, protein, fat, and BCAAs intakes were calculated by combining dietary data with Chinese food composition tables [27,28].

### 2.3. Laboratory Measurement

Blood samples drawn from the antecubital vein were collected from all subjects, and the blood samples were separated into plasma within 1 h, sent to the laboratory through the cold chain and frozen at −80 °C for later use. Fasting plasma glucose (FPG), serum uric acid (SUA), total cholesterol, triglycerides, LDL-C, HDL-C were detected using Hitachi Autoanalyzer 7600 (Hitachi, Tokyo, Japan). All measurements are performed by professional laboratory personnel, while strict quality control is carried out in the laboratory.

### 2.4. Statistical Analysis

Basic characteristics of the case and control groups were compared using Student’s t-test (continuous variable) and χ^2^ test (categorical variable). Generalized linear models were used to estimate the association between blood lipids and dietary BCAAs. Multiple logistic regression analysis was used to estimate the association between dietary BCAAs and risk of dyslipidemia. To estimate the odds ratios (ORs) for different types of dyslipidemia, dietary BCAAs are grouped by quartile: quartile 1 (Q1), <7.03 g/day; quartile 2 (Q2), 7.03 to <9.64 g/day; quartile 3 (Q3), 9.64 to <13.09; quartile 4 (Q4), ≥13.09. Regression models were adjusted for potential confounders, including age, sex, region, BMI, protein intake (g/day), carbohydrate intake (g/day), fat intake (g/day), current smoking status (yes/no), current alcohol consumption (yes/no), metabolic equivalents (MET-h/day) and education level (none or primary, middle, high school or college). Since the overall sample size is close to 20,000, according to the central limit theorem, the distributions of various variables were close to the normal distribution. In terms of sensitivity analysis, we performed stratified analysis by gender, age, region, BMI, smoking, alcohol consumption and metabolic equivalent. Meanwhile, interaction analysis was performed to evaluate the effect of stratification factors on the relationship between BCAAs and risk of dyslipidemia. We further explored the nonlinear relationship between dietary BCAAs and ORs of dyslipidemia using Restricted cubic splines (RCS) and selected 3 nodes for curve fitting according to the AIC optimality principle. Statistical analysis was performed using R 4.1.2. *p* < 0.05 (two-tailed) was considered to be significant.

## 3. Results

Table 1 presents the demographic and clinical characteristics of dyslipidemia and control groups. Compared with the control group, the systolic blood pressure (SBP), diastolic blood pressure (DBP) and dietary BCAAs intakes, levels of education, smoking rate and alcohol consumption rate in dyslipidemia group was significantly higher (*p* < 0.05). In addition, the dyslipidemia group had lower energy intake, fat intake, metabolic equivalents. As expected, FPG, SUA, TC, TG, LDL-C were higher and HDL-C was lower in dyslipidemia cases compared with controls (*p* < 0.001). The radar chart in Figure 2 visualizes the differences in blood parameters between the case group and the control group, which is consistent with the results in Table 1. The histogram in Figure 3 shows the distribution of dietary BCAAs in different genders of the two groups. The red dotted line represents the respective medians. It can be seen that the intake of dietary BCAAs in males was significantly higher than that in females (t = 27.54, *p* < 0.0001). Meanwhile, a total of 546 subjects in the case group were taking lipid-lowering medication.

Table 2 explores the linear relationship between dietary BCAAs and blood lipids in different groups. In the case and control groups, dietary BCAAs were positively correlated with TC and LDL-C (*p* < 0.05). However, TG was negatively correlated with dietary BCAAs in the control group (*p* = 0.0029), but not in the case group (*p* > 0.05). Additionally, HDL-C fluctuated with dietary BCAAs in each group (*p* < 0.05). 

Table 3 presents the logistic regression results for association between dietary BCAAs and risk of all types of dyslipidemia. After the model was adjusted for confounding factors, such as age, sex and BMI, region, current smoking status, current alcohol consumption, metabolic equivalents (MET-h/d), carbohydrate intake, protein intake, fat intake and educational level, ORs (95% CI) were statistically significant for Hypercholesteremia and Hyper-LDL-cholesterolemia. Higher dietary BCAAs were associated with higher ORs of Hypercholesteremia. The OR (95% CI) for Hypercholesteremia was 1.29 (1.05, 1.58), comparing the highest and lowest quartiles of dietary BCAAs intake. The ORs (95% CI) of Hyper-LDL-cholesterolemia were only statistically significant in the Q2 and Q3 segments, which were 1.15 (1.01, 1.31) and 1.20 (1.03, 1.39), respectively.

We focused sensitivity analysis on Hypercholesteremia in Table 4. The results showed that the positive association of dietary BCAAs with risk of Hypercholesteremia was almost consistent across all stratified analyses. The association appeared to be more stable in subjects in the lower age group or the lower energy intake group. In addition, age had an interactive effect on this relationship (*p* = 0.0047).

Table 5 shows dietary sources of dietary BCAAs in dyslipidemia and normolipemic groups. The top six categories of foods that contribute to dietary BCAAs were cereals, red meat, vegetables, fish and seafoods, beans and eggs.

In the RCS based on logistic regression model, the ORs of Hypercholesteremia increased significantly with dietary BCAAS when dietary BCAAs were below 10 g/day, followed by a slower rate of increase (Figure 4). The nonlinear spline test was statistically significant (*p* nonlinearity = 0.0059), indicating a potential nonlinear relationship between dietary BCAAs and risk of Hypercholesteremia. The same phenomenon was observed in 3 BCAAs (Ile, Leu and Val) separately (Figure 5). In addition, with the increasing of dietary BCAAs, the ORs of Hyper-LDL-cholesterolemia followed an inverted U-shaped trend (Figure 6), but the ORs were statistically significant in the range of dietary BCAAs 8–18 g/day. The nonlinear spline test is also statistically significant (*p* nonlinearity = 0.0198)*,* which is consistent with the trend of the results in Table 3.

## 4. Discussion

To the best of our knowledge, this is the first large-scale population-based study for the association between dietary BCAAs and risk of various types of dyslipidemia. In our study, dietary BCAAs were found to be positively correlated with TC and LDL-C. Meanwhile, higher dietary BCAAs were associated with higher ORs of Hypercholesteremia. Dietary BCAAs (range Q2 to Q3) were risk factors for Hyper-LDL-cholesterolemia (OR > 1, *p* < 0.05). These results were relatively stable after adjustment for potential confounders. In addition, the relationships between dietary BCAAs and the risk of Hypercholesteremia and Hyper-LDL-cholesterolemia were non-linear.

Our study is the first to verify the association between dietary BCAAs and blood lipids in a population-based sample. In both case and control groups, dietary BCAAs were positively correlated with TC and LDL-C. TG was negatively associated with dietary BCAAs in the control group, but not statistically associated in the case group. In addition, HDL-C fluctuated with dietary BCAAs. Previous limited studies have shown that restriction of dietary BCAAs can effectively reduce TG storage in the heart of Zucker fat rats [29]. To date, there were only studies on the relationship between plasma BCAAs and lipid metabolism. In a cross-sectional study of Chinese population, plasma BCAA levels were positively correlated with small dense low-density lipoprotein cholesterol (sdLDL-C), remnant-like particle cholesterol (RLP-C), and TG levels, but negatively correlated with HDL-C [30]. In a study of United States, women with higher plasma BCAAs had lower HDL-C (49.0 vs. 55.0 mg/dL) and higher TG (143 vs. 114 mg/dL), LDL-C (133 vs. 124 mg/dL). All types of BCAAs (isoleucine, leucine, and valine) were similarly associated with these indicators [31]. In addition, several cross-sectional studies have confirmed that there is a correlation between visceral fat content and plasma BCAAs in obese subjects [32,33]. In healthy subjects, there was a positive correlation between plasma BCAAs and both visceral and subcutaneous fat [34]. In study of gene regulation, it was observed that BCAAs degradation is a biological pathway for serum HDL-C-related transcripts [35]. Considering the evidence from these previous studies, there was physiological basis on the association between dietary BCAAs and lipid. 

Our findings suggest that dietary BCAAs intake is positively associated with the risk of Hypercholesteremia. This relationship remained very stable after confounder adjustments, stratified analyses, and nonlinear model exploration. This finding extends the relationship between BCAAs and disease at the dietary level. At present, limited studies can be used for reference and analogy, focusing on plasma BCAAs. In Feng-Hua Wang’s study [30], the highest tertile of plasma BCAAs compared with the lowest tertile, the OR of sdLDL-C was 2.33 (95% CI: 1.35, 4.03), 3.63 (95% CI: 1.69, 7.80) for RLP-C and 3.10 (95% CI: 1.66, 5.80) for TG. Studies have also shown that serum BCAAs levels are not only associated with obesity and impaired glucose tolerance, but also with dyslipidemia [23,36], which are characterized by elevated serum TG and decreased HDL-C [37]. As the consensus of the medical community, dyslipidemia is a risk factor for cardiovascular disease (CVD) [38,39], whereas high serum BCAA levels have been proved to be predictors or markers for dyslipidemia [22,23]. These findings suggest that plasma BCAAs are risk factors for atherosclerosis. Coincidentally, catabolites of BCAAs have been shown to be independently associated with coronary artery disease [40]. On the basis of previous studies, our findings seem to construct a complete chain that dietary BCAAs can directly or indirectly (through BCAAs in plasma) affect the occurrence of blood lipids or dyslipidemia. Our study demonstrates that dietary BCAAs have the same potential to cause dyslipidemia as plasma BCAAs.

In the present study, sensitivity analysis showed that the positive association of dietary BCAAs with risk of Hypercholesteremia was almost consistent across all stratified analyses. Meanwhile, age has an interactive effect on this relationship. It indicated that the effect of dietary BCAAs on lipid metabolism is closely related to the individual’s physiological state. The reasons for this phenomenon are multifaceted, among which the possible reasons include gene expression pathways, the activities of metabolic enzymes, and the balance restriction of energy metabolism. We found from the scarce literature that low carbohydrate and calorie restriction diets resulted in lower BCAA biosynthesis in intestinal flora [41]. Taken together with our results, low calorie intake may have contributed to a greater sensitivity to relationship between dietary BCAAs and Hypercholesteremia.

The ORs of Hypercholesteremia in nonlinear exploration increased rapidly with increasing dietary BCAAs and then slowed down. The ORs of Hyper-LDL-cholesterolemia showed an inverted U-shaped relationship. One possible reason could be that the dietary BCAAs and circulating BCAAs might also be non-linear with a “plateau effect”, like dietary cholesterol and blood cholesterol [42]. Meanwhile, various indications implied that dietary BCAAs had a complex intertwined relationship with blood lipids and dyslipidemia. Regarding this mechanism, a study from Finland proposed a hypothesis that the catabolism of BCAA is intertwined with the tricarboxylic acid (TCA) cycle and lipid metabolism [24], and this hypothesis has been validated in older Finnish men. This partly proved that the reason for the nonlinear relationship was the intertwined relationship between serum BCAAs levels and lipid metabolism.

Our study has several strengths. First, we used stricter matching principles than general studies, namely sex, age (±2 years), and BMI (±3 kg/m^2^). Considering that BMI is a key factor affecting blood lipids [43,44], we added the BMI as matching variable. Second, we developed extremely stringent inclusion criteria to ensure that observations were not interfered by other diseases. All subjects suffering from hypertension, diabetes, coronary heart disease, stroke, chronic obstructive pulmonary disease, asthma, bone and joint disease, neck and shoulder disease, chronic digestive system disease, chronic urinary system disease, tumor and other diseases were excluded from the study. Third, strict quality control procedures. We have developed unified work manual and experimental manual and conducted unified training for all investigators. Meanwhile, national and provincial quality control groups have been formed to track and control each link. Fourth, we performed stratified analysis, interaction analysis and nonlinear analysis. Using these statistical methods, the dose–response relationship and stability of dietary BCAAs and blood lipids were greatly validated.

There are still several limitations in our research. First, our study was based on a case–control study design, so causal relationship cannot be established. Second, measurement errors are unavoidable due to using our dietary recall method: 24 h retrospective method. The 24 h retrospective method is based on subjects’ recall and therefore suffers from recall bias. Third, dietary cholesterol data are lacking in our study, which will be further refined in future studies.

## 5. Conclusions

Our study observed a positive correlation between dietary BCAAs and TC and LDL-C. Meanwhile, higher dietary BCAAs were associated with higher ORs of Hypercholesteremia. The relationship between dietary BCAAs and risk of Hypercholesteremia is non-linear. The causal relationship requires further validation in experimental studies or prospective cohorts.

## Figures and Tables

**Figure 1 nutrients-14-01824-f001:**
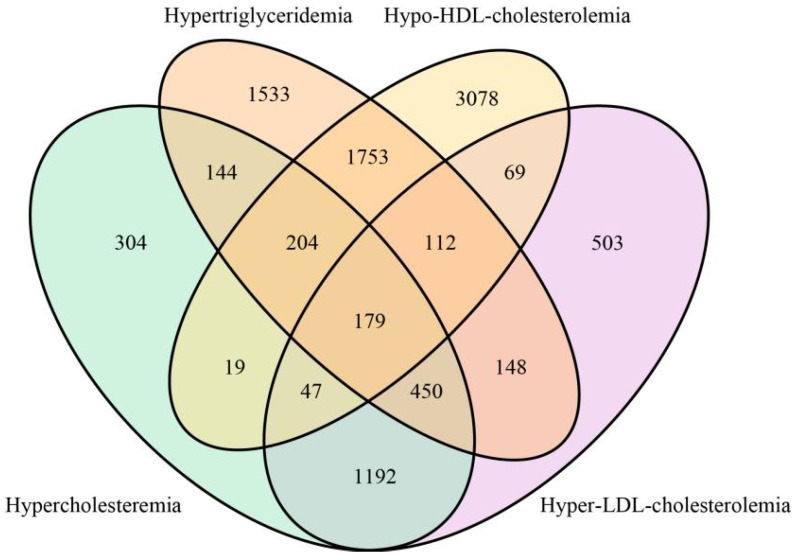
Venn diagram of sample distribution of four types of dyslipidemia subjects.

**Figure 2 nutrients-14-01824-f002:**
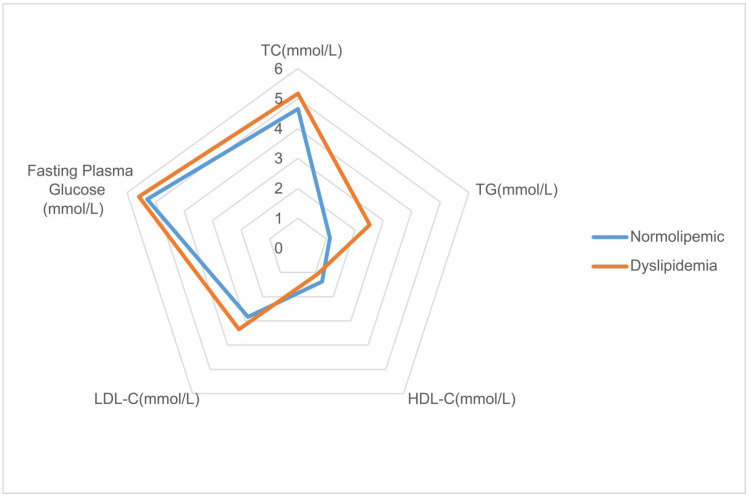
Radar plot of FPG and blood lipids in case group and control group.

**Figure 3 nutrients-14-01824-f003:**
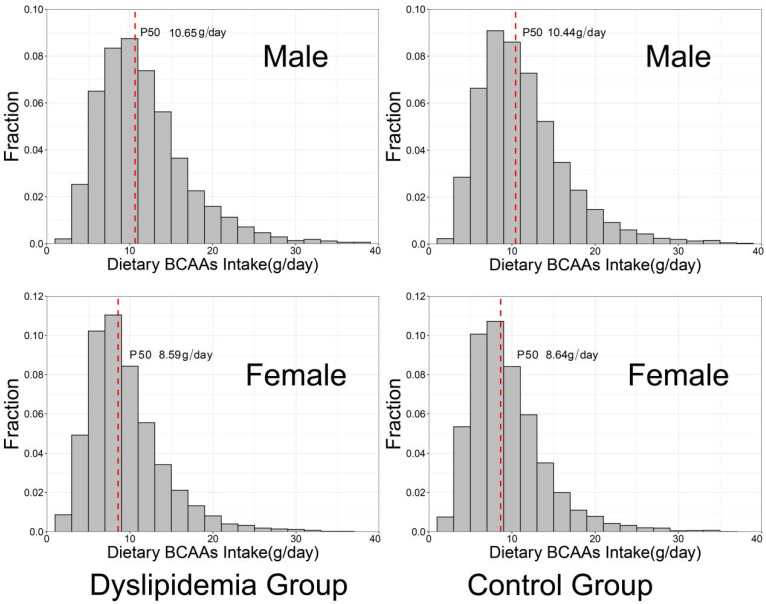
Distribution of dietary BCAAs consumption of different genders in case and control groups.

**Figure 4 nutrients-14-01824-f004:**
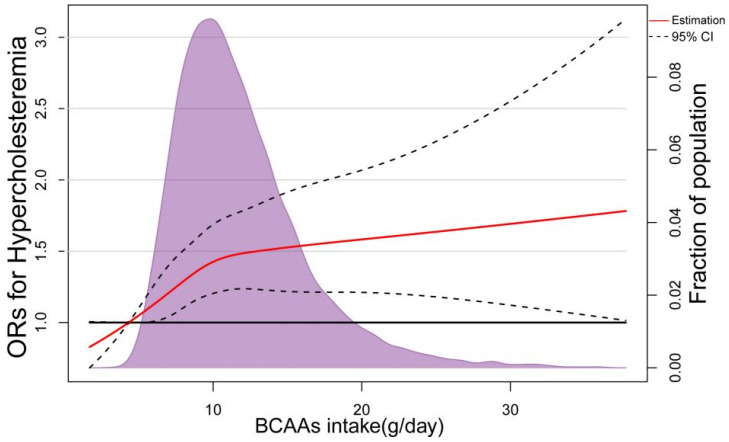
Representation of restricted cubic spline logistic regression models for dietary BCAAs and risk of Hypercholesteremia. Red Solid line, OR as a function of dietary BCAAs adjusted for age, sex and BMI, region, current smoking status, current alcohol consumption, metabolic equivalents (MET-h/day), carbohydrate intake, protein intake, fat intake and educational level; dashed lines, 95% CIs.

**Figure 5 nutrients-14-01824-f005:**
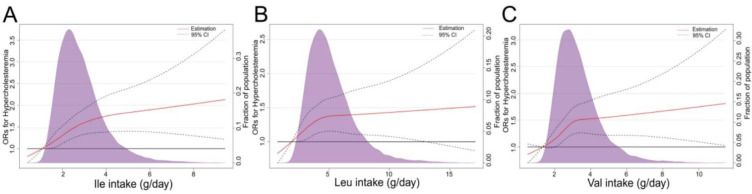
Representation of restricted cubic spline logistic regression models for three dietary BCAAs (Ile, Leu and Val) and risk of Hypercholesteremia. Red Solid line, OR as a function of specific BCAA adjusted for age, sex and BMI, region, current smoking status, current alcohol consumption, metabolic equivalents (MET-h/d), carbohydrate intake, protein intake, fat intake and educational level; dashed lines, 95% CIs. (**A**) RCS logistic regression models for dietary Ile and ORs of hypercholesterolemia. (**B**) RCS logistic regression models for dietary Leu and ORs of hypercholesterolemia. (**C**) RCS logistic regression models for dietary Val and ORs of hypercholesterolemia.

**Figure 6 nutrients-14-01824-f006:**
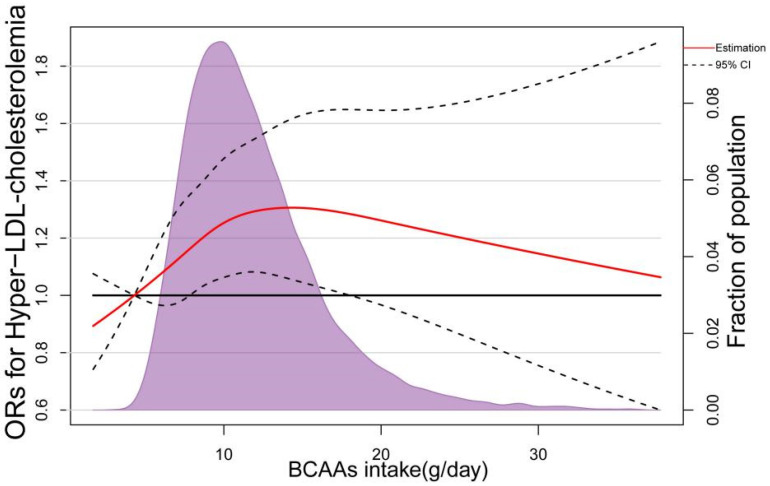
Representation of restricted cubic spline logistic regression models for dietary BCAAs and risk of Hyper-LDL-cholesterolemia. Red Solid line, OR as a function of dietary BCAAs adjusted for age, sex and BMI, region, current smoking status, current alcohol consumption, metabolic equivalents (MET-h/d), carbohydrate intake, protein intake, fat intake and educational level; dashed lines, 95% CIs.

**Table 1 nutrients-14-01824-t001:** Demographic and clinical characteristics of dyslipidemia and control groups.

	Hypercholesteremia (*n* = 2539)	Hypertriglyceridemia (*n* = 4523)	Hypo-HDL-Cholesterolemia (*n* = 5461)	Hyper-LDL-Cholesterolemia (*n* = 2700)	Dyslipidemia (*n* = 9792)	Normolipemic (*n* = 9541)	*p* Value
							Dyslipidemia vs. Normolipemic
Age (years)	55.51 (11.78)	51.61 (11.69)	51.01 (12.56)	55.24 (11.89)	52.45 (12.35)	52.69 (12.14)	0.181
Male, *n* (%)	1239 (48.8)	2700 (59.69)	3314 (60.68)	1352 (50.07)	5522 (56.39)	5284 (55.38)	0.157
BMI (kg/m^2^)	24.36 (22.87)	25.16 (29.53)	24.97 (26.93)	24.64 (22.18)	24.73 (25.96)	24.6 (12.04)	0.646
Fasting plasma glucose (mmol/L)	5.8 (1.98)	5.83 (2.04)	5.53 (1.67)	5.71 (1.83)	5.59 (1.73)	5.3 (1.19)	<0.001
Serum uric acid (μmol/L)	334.31 (95.36)	349.84 (95.59)	329.71 (91.61)	335.51 (93.13)	330.86 (92.72)	303.15 (81.01)	<0.001
Total cholesterol (mmol/L)	6.7 (3.43)	5.34 (2.7)	4.5 (1.07)	6.52 (0.8)	5.17 (2.12)	4.66 (0.69)	<0.001
Triglyceride (mmol/L)	2.84 (3.85)	3.89 (3.05)	2.71 (2.93)	2.05 (1.27)	2.53 (2.45)	1.12 (0.46)	<0.001
HDL cholesterol (mmol/L)	1.4 (0.43)	1.03 (0.28)	0.87 (0.14)	1.34 (0.38)	1.09 (0.36)	1.38 (0.28)	<0.001
LDL cholesterol (mmol/L)	4.34 (1.03)	3.32 (0.91)	2.85 (0.82)	4.6 (0.63)	3.35 (1.03)	2.84 (0.63)	<0.001
Systolic pressure (mmHg)	140.11 (31.36)	139.29 (35.98)	134.3 (28.32)	139.51 (30.84)	136.81 (31.79)	135.15 (30.21)	<0.001
Diastolic pressure (mmHg)	83.27 (26.78)	84.87 (33.84)	81.21 (24.31)	82.8 (26.19)	82.38 (28.29)	80.69 (26.42)	<0.001
Current smoker, *n* (%)	719 (28.32)	1530 (33.83)	1805 (33.05)	789 (29.22)	3071 (31.36)	2818 (29.54)	0.006
Current drinker, *n* (%)	349 (13.75)	680 (15.03)	662 (12.12)	355 (13.15)	1268 (12.95)	1117 (11.71)	0.009
Educational level							
None or elementary school	1254 (49.39)	1891 (41.81)	2162 (39.59)	1360 (50.37)	4300 (43.91)	4630 (48.53)	<0.001
Middle school	747 (29.42)	1578 (34.89)	1936 (35.45)	825 (30.56)	3282 (33.52)	3102 (32.51)	
High school	356 (14.02)	655 (14.48)	822 (15.05)	358 (13.26)	1394 (14.24)	1206 (12.64)	
College	182 (7.17)	399 (8.82)	541 (9.91)	157 (5.81)	816 (8.33)	603 (6.32)	
Energy intake (kcal/day)	1764.41 (614.1)	1822.73 (636)	1808.29 (612.29)	1757.15 (611.38)	1798.21 (621.87)	1822.79 (630.62)	0.006
Carbohydrate intake (g/day)	220.68 (88.74)	233.96 (94.11)	241.04 (98.02)	221.8 (89.11)	233.85 (95.25)	235.05 (96.56)	0.387
Protein intake (g/day)	55.59 (24.44)	56.36 (24.75)	56.12 (24.21)	55.65 (24.58)	55.77 (24.43)	54.71 (23.5)	0.002
Fat intake (g/day)	74.36 (40.24)	74.83 (40.89)	72.07 (38.74)	73.92 (40)	73.3 (39.74)	75.58 (40.77)	<0.001
MET-h/d	22.61 (17.97)	22.07 (16.96)	21.42 (16.74)	22.71 (17.70)	21.95 (17.24)	23.62 (19.09)	<0.001
BCAAs intake (g/day)	10.83 (5.23)	10.8 (5.21)	10.74 (5.17)	10.76 (5.13)	10.73 (5.19)	10.57 (5.15)	0.029
Ile intake (g/day)	2.72 (1.31)	2.71 (1.31)	2.69 (1.29)	2.70 (1.29)	2.69 (1.30)	2.65 (1.31)	0.0700
Leu intake (g/day)	4.89 (2.40)	4.88 (2.40)	4.87 (2.38)	4.86 (2.34)	4.86 (2.39)	4.79 (2.39)	0.0706
Val intake (g/day)	3.22 (1.53)	3.20 (1.52)	3.19 (1.52)	3.20 (1.50)	3.18 (1.52)	3.15 (1.53)	0.0836

Data are given as the mean (SD) or *n* (%).

**Table 2 nutrients-14-01824-t002:** Blood lipids according to BCAAs consumption levels and Dyslipidemia status (mean values and standard deviations).

	Quartile of Dietary BCAAs Intake (g/Day)				
	Q1 (Referent), <7.03	Q2, 7.03 to <9.64	Q3, 9.64 to <13.09	Q4, ≥13.09	*p* for Trend
Total					
*n* (%)	4833 (25.0)	4833 (25.0)	4833 (25.0)	4834 (25.0)	
Total cholesterol (mmol/L)	4.86 (1.05)	4.93 (2.62)	4.93 (1.09)	4.95 (1.06)	<0.0001
Triglyceride (mmol/L)	1.78 (1.6)	1.82 (1.9)	1.86 (2.15)	1.88 (1.93)	0.265
HDL cholesterol (mmol/L)	1.24 (0.34)	1.23 (0.35)	1.23 (0.36)	1.22 (0.36)	<0.0001
LDL cholesterol (mmol/L)	3.05 (0.87)	3.08 (0.89)	3.10 (0.91)	3.15 (0.9)	<0.0001
Dyslipidemia Group					
*n* (%)	2403 (24.54)	2431 (24.83)	2445 (24.97)	2513 (25.66)	
Total cholesterol (mmol/L)	5.09 (1.28)	5.22 (3.61)	5.17 (1.32)	5.19 (1.26)	0.0007
Triglyceride (mmol/L)	2.43 (2.03)	2.5 (2.46)	2.59 (2.8)	2.6 (2.43)	0.9536
HDL cholesterol (mmol/L)	1.10 (0.35)	1.10 (0.36)	1.08 (0.35)	1.09 (0.37)	0.0014
LDL cholesterol (mmol/L)	3.31 (1.01)	3.34 (1.03)	3.34 (1.06)	3.39 (1.02)	<0.0001
Control Group					
*n* (%)	2430 (25.47)	2402 (25.18)	2388 (25.03)	2321 (24.33)	
Total cholesterol (mmol/L)	4.63 (0.69)	4.65 (0.69)	4.68 (0.7)	4.69 (0.69)	<0.0001
Triglyceride (mmol/L)	1.14 (0.45)	1.13 (0.46)	1.11 (0.46)	1.11 (0.46)	0.0029
HDL cholesterol (mmol/L)	1.37 (0.27)	1.37 (0.28)	1.39 (0.3)	1.37 (0.29)	<0.0001
LDL cholesterol (mmol/L)	2.80 (0.62)	2.83 (0.62)	2.85 (0.63)	2.88 (0.64)	0.0026

Data are given as the mean and SD for continuous variables.

**Table 3 nutrients-14-01824-t003:** ORs (95% CI) of four dyslipidemia types, by quartiles of dietary BCAAs consumption.

	Quartile of Dietary BCAAs Consumption (g/Day)				*p* for Trend
	Q1 (Referent), <7.03	Q2, 7.03 to <9.64	Q3, 9.64 to <13.09	Q4, ≥13.09	
Hypercholesteremia vs. Control group					
Case/control subjects, *n*	593/2430	623/2402	663/2388	660/2321	
Crude OR (95% CI)	1	1.06 (0.94, 1.20)	1.14 (1.01, 1.29)	1.17 (1.03, 1.33)	0.0565
Adjusted OR * (95% CI)	1	1.14 (1.01, 1.30)	1.28 (1.13, 1.45)	1.39 (1.22, 1.59)	<0.0001
Adjusted OR † (95% CI)	1	1.15 (1.00, 1.32)	1.25 (1.08, 1.46)	1.29 (1.05, 1.58)	0.034
Hypertriglyceridemia vs. Control group					
Case/control subjects, *n*	1097/2430	1107/2402	1128/2388	1191/2321	
Crude OR (95% CI)	1	1.02 (0.92, 1.13)	1.05 (0.95, 1.16)	1.14 (1.03, 1.26)	0.0646
Adjusted OR * (95% CI)	1	0.99 (0.89, 1.10)	0.99 (0.889, 1.09)	1.02 (0.92, 1.14)	0.8894
Adjusted OR † (95% CI)	1	0.95 (0.85, 1.05)	0.92 (0.81, 1.04)	0.90 (0.76, 1.06)	0.5309
Hypo-HDL-cholesterolemia vs. Control group					
Case/control subjects, *n*	1331/2430	1351/2402	1369/2388	1410/2321	
Crude OR (95% CI)	1	1.02 (0.93, 1.12)	1.05 (0.95, 1.15)	1.11 (1.01, 1.22)	0.1528
Adjusted OR * (95% CI)	1	0.98 (0.89, 1.08)	0.97 (0.88, 1.06)	0.97 (0.88, 1.07)	0.9034
Adjusted OR † (95% CI)	1	0.93 (0.84, 1.02)	0.89 (0.79, 1.00)	0.87 (0.74, 1.01)	0.2402
Hyper-LDL-cholesterolemia vs. Control group					
Case/control subjects, *n*	633/2430	677/2402	698/2388	692/2321	
Crude OR (95% CI)	1	1.09 (0.96, 1.23)	1.13 (1.00, 1.28)	1.15 (1.02, 1.30)	0.1145
Adjusted OR * (95% CI)	1	1.16 (1.02, 1.31)	1.24 (1.09, 1.40)	1.33 (1.17, 1.51)	0.0001
Adjusted OR † (95% CI)	1	1.15 (1.01, 1.31)	1.20 (1.03, 1.39)	1.18 (0.97, 1.45)	0.1013
Dyslipidemia vs. Control group					
Case/control subjects, *n*	2403/2430	2431/2402	2445/2388	2513/2321	
Crude OR (95% CI)	1	1.02 (0.94, 1.11)	1.04 (0.957, 1.12)	1.10 (1.01, 1.19)	0.1388
Adjusted OR * (95% CI)	1	1.02 (0.94, 1.10)	1.02 (0.939, 1.10)	1.06 (0.98, 1.16)	0.4982
Adjusted OR † (95% CI)	1	0.98 (0.90, 1.06)	0.96 (0.87, 1.06)	0.95 (0.828, 1.08)	0.8417

* Model 1, adjusted for age, sex and BMI. † Model 2, adjusted for Model 1, region, current smoking status, current alcohol consumption, metabolic equivalents (MET-h/d), carbohydrate intake, protein intake, fat intake, physical activity and educational level.

**Table 4 nutrients-14-01824-t004:** Stratified analyses of Hypercholesteremia risk and dietary BCAAs consumption.

	Quartile of Dietary BCAAs Consumption (g/Day)				*p* Value for Interaction
	Q1 (Referent), <7.03	Q2, 7.03 to <9.64	Q3, 9.64 to <13.09	Q4, ≥13.09	
Sex					0.8826
Male (6523)	1	1.17 (0.95, 1.42)	1.29 (1.06, 1.56)	1.35 (1.12, 1.63)	
Female (5557)	1	1.14 (0.96, 1.35)	1.29 (1.08, 1.53)	1.46 (1.21, 1.76)	
Age, years					0.0047
<55 (6845)	1	1.24 (1.02, 1.51)	1.50 (1.25, 1.82)	1.64 (1.36, 1.98)	
≥55 (5235)	1	1.10 (0.93, 1.30)	1.11 (0.93, 1.32)	1.18 (0.98, 1.42)	
BMI, kg/m^2^					0.0739
<24 (5215)	1	1.33 (1.10, 1.61)	1.43 (1.18, 1.74)	1.47 (1.20, 1.81)	
≥24 (6865)	1	1.03 (0.87, 1.22)	1.20 (1.02, 1.42)	1.34 (1.13, 1.58)	
Current smoking					0.7721
Yes (3537)	1	1.23 (0.95, 1.59)	1.21 (0.94, 1.57)	1.328 (1.04, 1.70)	
No (8543)	1	1.12 (0.97, 1.30)	1.32 (1.14, 1.53)	1.43 (1.22, 1.67)	
Current drinking					0.4331
Yes (1466)	1	0.89 (0.60, 1.32)	1.16 (0.80, 1.67)	1.17 (0.82, 1.66)	
No (10,614)	1	1.18 (1.03, 1.35)	1.29 (1.13, 1.48)	1.41 (1.23, 1.63)	
METs-h/day					0.3213
<22.77 (7439)	1	1.06 (0.90, 1.24)	1.10 (0.94, 1.29)	1.28 (1.09, 1.51)	
≥22.77 (4641)	1	1.35 (1.09, 1.69)	1.71 (1.37, 2.12)	1.67 (1.34, 2.08)	
Region					0.4076
urban	1	1.06 (0.86, 1.31)	1.32 (1.08, 1.62)	1.44 (1.17, 1.77)	
rural	1	1.20 (1.02, 1.41)	1.24 (1.06, 1.47)	1.36 (1.15, 1.61)	
Energy intake (kcal/day)					0.2401
<2200 (9325)	1	1.15 (1.01, 1.32)	1.34 (1.17, 1.54)	1.47 (1.26, 1.73)	
≥2200 (2755)	1	1.22 (0.62, 2.39)	1.34 (0.72, 2.48)	1.62 (0.89, 2.96)	

Data are OR (95% CI). The multivariate model was adjusted for age, sex and BMI.

**Table 5 nutrients-14-01824-t005:** Dietary sources of dietary BCAAs in dyslipidemia and normolipemic groups.

	Normolipemic (*n* = 9541)	Dyslipidemia (*n* = 9792)	Total (*n* = 19,333)
Cereals (g/day)	3.88 (2.25)	3.82 (2.23)	3.85 (2.24)
Beans (g/day)	0.78 (1.36)	0.75 (1.28)	0.77 (1.32)
Vegetables (g/day)	0.93 (1.13)	0.97 (1.15)	0.95 (1.14)
Fruits (g/day)	0.04 (0.26)	0.04 (0.25)	0.04 (0.26)
Nuts (g/day)	0.12 (0.44)	0.12 (0.46)	0.12 (0.45)
Red meat (g/day)	1.82 (2.09)	1.92 (2.20)	1.87 (2.14)
Poultry (g/day)	0.43 (1.08)	0.45 (1.12)	0.44 (1.10)
Dairy products (g/day)	0.13 (0.62)	0.12 (0.51)	0.12 (0.56)
Eggs (g/day)	0.55 (0.76)	0.57 (0.76)	0.56 (0.76)
Fish and seafoods (g/day)	0.85 (1.85)	0.98 (1.98)	0.92 (1.92)
Snacks (g/day)	0.24 (1.25)	0.2 (1.06)	0.22 (1.16)
Beverage (g/day)	0.18 (1.64)	0.18 (1.58)	0.18 (1.61)
Condiments (g/day)	0.13 (0.36)	0.14 (0.50)	0.13 (0.43)
Others (g/day)	0.47 (1.18)	0.47 (1.14)	0.47 (1.16)

## Data Availability

The data are not allowed to be disclosed according to the National Institute for Nutrition and Health, Chinese Center for Disease Control and Prevention.

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
