# Peer review of "Dietary Branched-Chain Amino Acids (BCAAs) and Risk of Dyslipidemia in a Chinese Population"

_nutrients, 2022, doi:10.3390/nu14091824_

Round 1

Reviewer 1 Report

Yu et al., conducted a case-control analysis to examine the dietary BCAAs and risk of dyslipidemia. They observed a positive association between BCAAs intake and hypercholesteremia risk, and the association was non-linear, reaching a plateau at higher BCAAs levels. This results is interesting, but I have several comments for authors to consider.

1. Lines 60-62, there are four outcomes. It seems that there are overlaps for these outcomes (some participants could have both hypercholesteremia and hyper-LDL-cholesterolemia). Please report these overlaps.
2. Lines 69-71, are these inclusion criteria for the controls, since there is a criteria of "normal lipid level in clinical examination and blood tests during this survey"?
3. Lines 96-97, has the three-day 24-hour retrospective food record method been validated?
4. Line 121-124, for the cases, did any of them receive lipid-lowering medication? Please include this information. Also, I think dietary cholesterol and total energy intake should also be adjusted.
5. Results section. I don't think this manuscript needs to include results of FPG and SUA. They are not quite related to the aim and could distract readers.
6. For BCAAs, is dietary intake of the three specific BCAAs available? If available, including the results for individual BCAAs will enhance the novelty of this paper.
7. Lines 186-187, 267-271, the interpretation for the subgroup analysis was not quite right. In table 4, the non-significant association among those with an energy intake >=2000 could be just due to the small sample size (2755 vs. 9325). The effect sizes for these two subgroups are generally similar, please do not overinterpret this.
8. Lines 265-267, this sentence is very vague and tells little things.
9. Lines 278-280, again, this sentence is also very vague and does not tell the reason for the nonlinear relationship. One possible reason could be that the dietary BCAAs and circulating BCAAs might also be non-linear with a "plateau effect", like the dietary cholesterol and blood cholesterol. 
10. Lines 296-297, I guess it's referring to the dietary record method, please specify it here. Also what kind of measurement error could this method have?
11. Lines 301-302, first, I think using 25th and 75th percentiles is difficult for generalizations; second, the results for hyper-LDL-cholesterolemia is not very significant, p for non-linearity is only 0.02. They can be removed from the conclusion.  

Author Response

Comments and Suggestions for Authors

Yu et al., conducted a case-control analysis to examine the dietary BCAAs and risk of dyslipidemia. They observed a positive association between BCAAs intake and hypercholesteremia risk, and the association was non-linear, reaching a plateau at higher BCAAs levels. This results is interesting, but I have several comments for authors to consider.

  1. Lines 60-62, there are four outcomes. It seems that there are overlaps for these outcomes (some participants could have both hypercholesteremia and hyper-LDL-cholesterolemia). Please report these overlaps.

Thanks for your constructive suggestions. We have supplemented the overlap in detail.

  1. Lines 69-71, are these inclusion criteria for the controls, since there is a criteria of "normal lipid level in clinical examination and blood tests during this survey"?

Thank you very much for your meticulous review. Yes, and the inclusion criteria of the case group were also Age ≥ 30 years old, BMI<40 kg/m2. We have made additional explanations in the text.
3. Lines 96-97, has the three-day 24-hour retrospective food record method been validated?

Thank you very much for your professional question. The three-day 24-hour retrospective food record method is a classic and commonly used method in nutritional epidemiological investigations. This approach has been used in many classic studies (1,2). So far, this method is in parallel with the food frequency questionnaire (FFQ) method, and has more advantages than FFQ in calculating energy and nutrients.

  • He Y, Li Y, Yang X, Hemler EC, Fang Y, Zhao L, Zhang J, Yang Z, Wang Z, He L, Sun J, Wang DD, Wang J, Piao J, Liang X, Ding G, Hu FB. The dietary transition and its association with cardiometabolic mortality among Chinese adults, 1982-2012: a cross-sectional population-based study. Lancet Diabetes Endocrinol. 2019 Jul;7(7):540-548. doi: 10.1016/S2213-8587(19)30152-4. Epub 2019 May 10. PMID: 31085143; PMCID: PMC7269053.

(2) Liu M, Liu C, Zhang Z, Zhou C, Li Q, He P, Zhang Y, Li H, Qin X. Quantity and variety of food groups consumption and the risk of diabetes in adults: A prospective cohort study. Clin Nutr. 2021 Dec;40(12):5710-5717. doi: 10.1016/j.clnu.2021.10.003. Epub 2021 Oct 11. PMID: 34743048.

4.Line 121-124, for the cases, did any of them receive lipid-lowering medication? Please include this information. Also, I think dietary cholesterol and total energy intake should also be adjusted.

Thanks for your professional advice. A total of 546 subjects in the case group were taking lipid-lowering medication. We have added this in the result.

In the statistical analysis, dietary carbohydrates, protein, and fat have been adjusted. Almost all dietary energy comes from these three substances. On second thought, we believe that it is not advisable to incorporate dietary energy into the model, which will lead to inevitable multicollinearity and reduce the reliability of the model.

At the same time, due to the shortcomings of our database, we do not have dietary cholesterol data, we are very sorry, and hope that we can further improve this part in future research.

  1. Results section. I don't think this manuscript needs to include results of FPG and SUA. They are not quite related to the aim and could distract readers.

Thank you for your professional opinion. We have deleted the corresponding sections in Table 2 and the results. At first we thought that there was an association between FPG, SUA and blood lipids. Describing their basic situation enables the reader to learn more about the basic situation of the subjects. So we still keep a small amount of FPG and SUA information in the basic information section.
6. For BCAAs, is dietary intake of the three specific BCAAs available? If available, including the results for individual BCAAs will enhance the novelty of this paper.

Thank you very much for your constructive suggestions. We have added dietary intake of the three specific BCAAs in Table 1 and marked them in red.
7. Lines 186-187, 267-271, the interpretation for the subgroup analysis was not quite right. In table 4, the non-significant association among those with an energy intake >=2000 could be just due to the small sample size (2755 vs. 9325). The effect sizes for these two subgroups are generally similar, please do not overinterpret this.

Thank you very much for your professional and detailed advice. We have revised it according to your comments.
8. Lines 265-267, this sentence is very vague and tells little things.

Thank you for your careful and responsible review. We have made additional changes to this part and marked in red.
9. Lines 278-280, again, this sentence is also very vague and does not tell the reason for the nonlinear relationship. One possible reason could be that the dietary BCAAs and circulating BCAAs might also be non-linear with a "plateau effect", like the dietary cholesterol and blood cholesterol. 

Thanks for your inspiring comments. We are more than happy to revise and supplement this section. Based on your suggestion, we have supplemented this section and marked it in red.
10. Lines 296-297, I guess it's referring to the dietary record method, please specify it here. Also what kind of measurement error could this method have?

Thanks for your detailed comments. Yes, it's referring to the dietary record method. We have made additional notes and marked them in red. The main cause of measurement error is recall bias.
11. Lines 301-302, first, I think using 25th and 75th percentiles is difficult for generalizations; second, the results for hyper-LDL-cholesterolemia is not very significant, p for non-linearity is only 0.02. They can be removed from the conclusion.  

Thank you very much for your professional insights. We are more than happy to adopt your suggestion. We have revised it according to your comments.

Reviewer 2 Report

Interesting paper, minor language revisions are needed

Author Response

Thank you very much for your review and your affirmation. We made some changes to the paper and marked it in red. Thank you for your review. Best wishes!

Reviewer 3 Report

The reviewer acknowledges authors' work to create a manuscript well written and with important results for a specific population that can be translated to the world population taking in account the specificities of the population analyzed.

Author Response

(The authors gave the same response as above.)

Round 2

Reviewer 1 Report

I have a few minor comments left:

Lines 64-74
Thanks for including these information. But could authors draw a Venn diagram to visualize the overlaps?

Lines 287-289
"Plateau effect" is just my thought for authors' consideration. If authors would like to include it in the discussion, please also add reference for that (check appendix 12 in this paper: https://www.sciencedirect.com/science/article/pii/S2213858717302838?casa_token=r1P-fo7Gl4oAAAAA:xQOZAIjHmDJ5nPqcgqcU_quHQFulodlb2IM5ygOGcZ-x8xnsla3x4_LTJzA6LsSa7jVWkHWIkzU#sec1). It does not need to dietary cholesterol, but could also be other nutrients. Such "Plateau effect" could explain the nonlinear relationship in Figure 3. For the U shape in Figure 4. I also don't have a clear idea. 

Lines 312-314
Just a correction for my typo in the comments during the first round: it's dietary recall, not dietary record. Also, lack of dietary cholesterol should be added as a limitation as well.

For specific BCAA variables, could also authors conduct an analysis for hypercholesteremia and  include all these three BCAAs in the same model, to check which BCAA is driving the association?

Could also provide a table showing the top food contributors to BCAA in this population?

Author Response

Lines 64-74

Thanks for including these information. But could authors draw a Venn diagram to visualize the overlaps?

Thanks for your fascinating suggestion, we've added Venn diagrams in the paper. This greatly improves the visualization of data. Thanks again for your guidance.

Lines 287-289

"Plateau effect" is just my thought for authors' consideration. If authors would like to include it in the discussion, please also add reference for that (check appendix 12 in this paper: https://www.sciencedirect.com/science/article/pii/S2213858717302838?casa_token=r1P-fo7Gl4oAAAAA:xQOZAIjHmDJ5nPqcgqcU_quHQFulodlb2IM5ygOGcZ-x8xnsla3x4_LTJzA6LsSa7jVWkHWIkzU#sec1). It does not need to dietary cholesterol, but could also be other nutrients. Such "Plateau effect" could explain the nonlinear relationship in Figure 3. For the U shape in Figure 4. I also don't have a clear idea. 

Thank you very much for your thoughtful and constructive suggestions. We have added references as you suggested. However, due to our original database, more refined nutrients cannot be added as adjustment factors. The results of our research are now more of a phenomenon. The underlying principles and mechanisms still need further research to explore. We will use your helpful suggestions as a reference for our further research. Thanks again for your inspiring advice and guidance.

Lines 312-314

Just a correction for my typo in the comments during the first round: it's dietary recall, not dietary record. Also, lack of dietary cholesterol should be added as a limitation as well.

For specific BCAA variables, could also authors conduct an analysis for hypercholesteremia and  include all these three BCAAs in the same model, to check which BCAA is driving the association?

Could also provide a table showing the top food contributors to BCAA in this population?

Thank you very much for your valuable advice. We have revised the typo again as you suggested, while adding a limitation that lacks dietary cholesterol data. Meanwhile, we have conducted an analysis for hypercholesteremia and include all these three BCAAs in the same model (Figure 5). Analysis results show that three BCAAs are similarly associated with hypercholesteremia, and we have explained that in the text. Finally, we have provided a table showing the food contributors to BCAAs in this population. Thanks again for your helpful advice.